# The Threat Posed by Environmental Contaminants on Neurodevelopment: What Can We Learn from Neural Stem Cells?

**DOI:** 10.3390/ijms24054338

**Published:** 2023-02-22

**Authors:** Raj Bose, Stefan Spulber, Sandra Ceccatelli

**Affiliations:** Department of Neuroscience, Karolinska Institutet, 171 77 Stockholm, Sweden

**Keywords:** developmental neurotoxicity, neural stem cells (NSCs), neurogenesis, molecular mechanisms, epigenetic modifications, methylmercury, per- and polyfluorinated substances (PFAS)

## Abstract

Exposure to chemicals may pose a greater risk to vulnerable groups, including pregnant women, fetuses, and children, that may lead to diseases linked to the toxicants’ target organs. Among chemical contaminants, methylmercury (MeHg), present in aquatic food, is one of the most harmful to the developing nervous system depending on time and level of exposure. Moreover, certain man-made PFAS, such as PFOS and PFOA, used in commercial and industrial products including liquid repellants for paper, packaging, textile, leather, and carpets, are developmental neurotoxicants. There is vast knowledge about the detrimental neurotoxic effects induced by high levels of exposure to these chemicals. Less is known about the consequences that low-level exposures may have on neurodevelopment, although an increasing number of studies link neurotoxic chemical exposures to neurodevelopmental disorders. Still, the mechanisms of toxicity are not identified. Here we review in vitro mechanistic studies using neural stem cells (NSCs) from rodents and humans to dissect the cellular and molecular processes changed by exposure to environmentally relevant levels of MeHg or PFOS/PFOA. All studies show that even low concentrations dysregulate critical neurodevelopmental steps supporting the idea that neurotoxic chemicals may play a role in the onset of neurodevelopmental disorders.

## 1. Introduction

Neurodevelopmental disorders, such as intellectual disabilities, attention-deficit/hyperactivity disorder (ADHD), dyslexia, autism spectrum disorder (ASD), as well as schizophrenia, bipolar disorder, and depression, are increasing globally causing immense suffering and huge costs to society [1]. Several of these pathological conditions may have a genetic origin, but an increasing number of studies suggest that other factors, including maternal stress, infections, malnutrition, low birth weight, and exposure to toxicants, may play a role in their etiopathogenesis [2]. More than 200 chemicals belonging to different classes, including metals, persistent organic pollutants, organic solvents, and pesticides, have been identified as neurotoxicants [2]. Experimental and epidemiological data indicate that developmental exposure to certain environmental contaminants poses a threat to the developing brain and may lead to neurodevelopmental disorders.

Specific and well-organized cellular and molecular processes, including proliferation, migration, differentiation, myelination, and synaptic pruning, characterize the development of the nervous system. Any incident disrupting the sequence of developmental steps can lead to permanent or transient structural and functional losses. The impact that harmful stimuli, such as neurotoxicants, exert on the developing nervous system depends on the timing of exposure that may coincide with region-specific windows of susceptibility. Neurodevelopmental damages may not be evident for a long time (silent neurotoxicity) up until different challenges, including aging, trauma, or exposure to toxicants, disclose and even build up on the existing cellular or biochemical damage. In addition, epigenetic modifications may occur, making the alterations heritable (see [3]).

We are exposed to a large and increasing number of chemicals and the lack of information regarding their neurodevelopmental toxic potential has generated a growing concern for public health. Experimental animal models and tests have been critical in toxicology to provide fundamental information about potential neurotoxicants. However, animal studies have intrinsic limitations, including variations across species, labor- and time-intensive procedures, as well as ethical issues [4,5]. Therefore, in vitro assays with cell lines and primary cells, especially those of human origin, have turned out to be good alternatives to live animal experiments for developmental neurotoxicity studies [6].

Chemical neurotoxicity investigations using neural stem cells (NSCs) have shown that the use of these cells provides unique information for the identification of neurodevelopmental toxicants and their mechanism of action. Here, we review original articles on mechanistic data generated using NSCs derived from mice, rats, and humans as experimental models. We specifically focus on the well-known neurotoxicant methylmercury (MeHg) and on two compounds belonging to the larger class of per- and poly-fluoroalkyl substances (PFAS), namely perfluorooctanesulfonic acid (PFOS) and perfluorooctanoic acid (PFOA).

## 2. Neural Stem Cells for Developmental Neurotoxicity Studies

NSCs are generated from the embryonic neuroectoderm, which eventually generates most cells in the central and peripheral nervous systems. The *Sox* gene family members and Otx2 are the earliest markers for NSCs, which are also known as neuroepithelial cells. The definitions NSCs and neural progenitor cells (NPCs) refer to undifferentiated cells with specific characteristics. NSCs are multipotent and have the ability to differentiate into neurons, astrocytes, and oligodendrocytes; they are self-renewing and maintain multipotency across an indefinite number of divisions. Instead, the potential of NPC is more restricted [7]. Both NSCs and NPCs are present throughout the development of the nervous system in the ventricular zone (VZ) and the subventricular zone (SVZ), and embryonic neurogenesis is essential for the formation of specific spatial organization, neuronal networking, and maturation of the nervous system. Signaling molecules in the microenvironment allow the maintenance, proliferation, and neuronal fate commitment of local stem cell populations throughout development. For experimental purposes, primary NSCs have been isolated from various regions of the embryonic/fetal nervous system, such as the olfactory bulb, subventricular zone, hippocampus, cerebellum, cerebral cortex, and spinal cord. In the adult rodent brain, NSCs located in the subventricular zone of the lateral ventricle and the subgranular layer of the hippocampal dentate gyrus are active and generate new neurons, astrocytes, and oligodendrocytes continuously throughout life. These newly generated neural cells play a crucial role in the maintenance of learning and memory [8]. The persistence and the ability to produce new neurons have not been fully clarified in the adult human brain.

The integration of newly generated neurons into pre-existing neuronal networks is essential for the function and the plasticity of neural circuits. Recent evidence showed that adult brain injuries induce the generation of cells characterized initially as specialized astrocytes. However, when cultured, these cells demonstrate NSC properties [9], such as multipotency and self-renewal. Thus, toxicity studies using NSCs are highly relevant not only for the developing brain but also for the adult nervous system. We briefly discuss below several NSC models that have been used to investigate known and suspected neurotoxicants.

### Cell Lines and Primary Neural Stem Cells Derived from Mice, Rats, and Humans

The C17.2 cells, from an immortalized NSC line derived from a murine neonatal cerebellum, have been widely used for understanding cell fate and differentiation of neural progenitors. Although these cells are transformed and have restrictions to generate functional neurons, they maintain the capacity to follow developmental cues. For example, they can generate fully functional neurons when transplanted into the mid-embryonic mouse brain during development but not at later developmental stages, when gliogenesis is predominant.

Primary cultures of NSCs have been successfully derived from the telencephalon, striatum, and hippocampus of rat embryos and retain multipotential properties [10]. These cells can be cultured as a monolayer (2D) on coated surfaces, or as three-dimensional (3D) on non-coated surfaces; the stemness is maintained by the addition of fibroblast growth factor (FGF), and/or epidermal growth factor (EGF). Upon removing EGF and FGF from the culture medium, the cells can differentiate into the major cell types found in the cerebral cortex, striatum, and hippocampus, including pyramidal and interneurons, astrocytes, and oligodendrocytes, as well as smooth muscle cells.

Primary cultures of adult NSCs (aNSCs) can be obtained from the anterior portion of the lateral walls of the lateral ventricles of adult rats [11]. These cells are cultured as 3D on non-coated surfaces in the presence of EGF for generating neurospheres within a week. Thereafter, mechanical or enzymatic dissociation of neurospheres produces single cells or smaller size spheres, which are either passaged for de novo neurosphere propagation or plated onto coated surfaces for monolayer culture, and further differentiated into neurons, astrocytes, and oligodendrocytes in the absence of EGF.

Human NPCs (hNPCs) have been produced from the fetal forebrain at gestational weeks 6–12 [11,12], cultured in the presence of FGF and EGF, and maintained in suspension as proliferating neurospheres [13]. When hNPCs are cultured on coated surfaces as a monolayer or as neurospheres without growth factors, they give rise to the major cell types found in the adult brain.

Human NSCs generated from umbilical cord blood (HUCB-NSC cell line) can differentiate into mature neurons expressing functional voltage- and ligand-gated ion channels and are capable of establishing functional networks. In addition, HUCB-NSC cells can also generate major glial cell lineages, such as astrocytes and oligodendroglia [14].

Human iPSC-derived neuroepithelial-like stem cells, such as the AF22 cell line and NES cells are cultured on a coated plate as a monolayer in the presence of EGF and FGF for propagation. They are characterized by the long-term self-renewing capacity and the rosette-like growth pattern after 8–12 days in the presence of growth factors. These cells give rise to neurons and glia when they are induced to differentiate by growth factor withdrawal. In addition, they retain neuro- and gliogenic potential even after long-term proliferation [15].

A recent advance in human iPSC technology is the generation of brain organoids that recapitulate complex processes occurring during embryonic development, and express cellular diversity, networking, and compartmentalization. A further step forward is the generation of vascularized brain organoids that offer unprecedented possibilities to understand the complexity of nervous system damage and disease development [16].

## 3. The Selected Developmental Neurotoxic Chemical Contaminants

In our daily lives, we are all exposed to a variety of chemicals present in products we use, in the food that we eat, or in our inner or outer environment (Figure 1). Exposure to chemicals poses a high risk to vulnerable groups including pregnant women, children, and the elderly. In particular, exposure during critical periods of development in prenatal and early life stages can predispose to disease later in adult life. The three neurotoxic chemical contaminants that we consider in this review are among the ones detected in human blood at levels that vary depending on geographic location, diet, and working environment. They can cross the placenta barrier, thus being of particular concern for developing organisms.

### 3.1. Methylmercury

The global environmental and food contaminant MeHg is mostly generated from inorganic mercury by the methylating activity of anaerobic bacteria in water sediments [17]. Contaminated aquatic food is the main source of MeHg exposure for human populations [17].

Several mechanisms of toxicity have been identified as responsible for its toxic effects including interactions with sulfhydryl groups of thiol-containing compounds, thereby targeting peptides and proteins containing cysteine and methionine; mitochondrial function impairment; perturbation of intracellular Ca^2+^ homeostasis; increased generation of reactive oxygen species (ROS) [18]. All these intracellular alterations have harmful effects on the nervous system, particularly during development.

The nervous system is known to be the major MeHg target organ in both animals and humans particularly during prenatal life. By crossing the placenta, as well as the blood-brain barrier, MeHg induces adverse effects on different entities depending on the time and duration of exposure [19]. Fetuses can be heavily affected even when mothers do not show signs of toxicity, which can be explained by the fact that MeHg-fetal blood levels are about 1.7–1.9 times higher than those measured in maternal blood [20].

The developmental neurotoxic effects of MeHg were first identified in the 1950s in Japan. Wastewater contaminated with mercury had been discharged by a chemical factory into Minamata Bay for decades. The accumulated mercury entered the aquatic food chain as MeHg and contaminated the local populations that had a fish-based diet. The most severe neurotoxic effects on humans were observed in children of women who had eaten MeHg-contaminated fish during pregnancy. Surviving children exhibited various neurological clinical signs including ataxia, blindness, spasticity, impairment of motor skills, and variable degrees of mental retardation depending on the severity of the prenatal exposure [21].

Thanks to major efforts to reduce the release of mercury in the environment, the MeHg aquatic contamination has been considerably reduced [22]. Nevertheless, the adverse long-term consequences that developmental exposure to low levels of MeHg (via the maternal diet) may have on the nervous system are a matter of great worry (see [23]). Experimental animal studies and epidemiological investigations of seafood-eating populations have identified behavioral alterations and decreased cognitive abilities linked to exposure to prenatal MeHg (see [24]). Currently, the general population gets exposed to MeHg by eating contaminated fish seafood, and marine mammals (Figure 1), and the concentration of MeHg in cord blood directly correlates with maternal fish intake even at low maternal fish consumption [22]. Therefore, in several countries, women who are pregnant or planning to become pregnant, nursing mothers, and young children are recommended to exclude the consumption of sea food containing high levels of MeHg [25].

The concentration of MeHg In the umbilical cord blood at birth provides an estimation of the level of exposure during development. Levels in the range of 1.5–3.5 µM can be detected in populations from areas exposed to heavy industrial pollution, and such high MeHg levels are mostly due to manmade outbreaks [22]. In most populations, the average concentration of MeHg in the cord blood is below 10 nM (~2.2 µg/L) and reaches 100 nM (~21.5 µg/L) if the diet is based primarily on seafood (see [26,27]). A safety maximum limit for cord blood was defined at 5.8 µg/L (~27 nM) [28]. Therefore, there is genuine concern regarding the potential developmental neurotoxic effects of MeHg.

### 3.2. Polyfluoroalkylated Substances

PFAS are synthetic chemicals used in commercial and industrial products, including water and oil repellants for paper, packaging, textile and leather goods, industrial surfactants, fire-fighting foams, food packaging, and non-stick cookware, due to their unique physicochemical characteristics [29,30,31,32,33,34]. While PFOS and PFOA are banned in the European Union and the United States, the most common PFAS contaminants are still found in the environment because of the long-term persistence of those chemicals.

Currently, the main source of exposure to PFAS, such as PFOS and PFOA in the general population, is via intake of contaminated foods, beverages, drinking water, and inhalation (Figure 1). Infants are exposed to these chemicals via breastfeeding while fetuses are exposed via maternal cord blood since both PFOS and PFOA can pass through the placenta as well as the blood-brain barrier (BBB). For example, evidence suggested that longer periods of breastfeeding resulted in higher blood PFAS levels in the infant [35]. In the USA, at least six million people can be exposed to PFOS and PFOA at a level of 70 ng/L [36], which is higher than the levels of EPA health advisories.

A recent cohort study reported that human maternal serum concentration of PFOS (4.4–6.0 ng/mL) and PFOA (11.2–15.6 ng/mL) [37] and prenatal exposure to PFOS was significantly associated with hyperactivity and hyperactive-type ADHD in young school-aged children (mean PFOS concentration in maternal serum 12.8 ng/mL) [37]. Another recent study reported that prenatal exposure to PFOA was associated with an increased risk of ASD and ADHD in children where the concentration of PFOA is in the range of 1.47–2.17 ng/mL in maternal plasma [38].

Several neurotoxic mechanisms have been proposed, but three have received particular attention: disruption of Ca^2+^ homeostasis; interference with neurotransmitter signaling; and neuroendocrine disruption (reviewed in [39]). Given the developmental neurotoxicity potential and the persistence in the environment, PFAS exposure is a matter of major concern.

## 4. Neurodevelopmental Toxicity in NSCs

The neurotoxic outcomes of exposure to environmental contaminants, such as apoptosis or alterations in proliferation and differentiation, can have several mechanisms, some of which are shared by different compounds. The identification of key events at cellular and molecular levels is critical to provide support for plausible adverse nervous system outcomes and also to indicate possible preventive and therapeutic strategies. In the following sections, we will focus on neurotoxic endpoints and underlying mechanisms separately. The main findings of the studies reviewed here are summarized in Table 1 (MeHg) and Table 2 (PFOS or PFOA).

### 4.1. Apoptosis

Very low toxic effects (IC5) for cell viability have been reported following exposure to 10 nM MeHg for 10 days, and 100 nM for 5 days in human NPCs derived from H9 human embryonic stem cells [40]. The lowest concentration inducing alterations in cell viability (LOAEC) after 24 h exposure to MeHg was 3000 nM for neurospheres generated from hiPSC-derived hNPCs, and 1000 nM in neurospheres derived from primary hNPCs, while EC50 was similar in the two models [41]. In hiPSC-derived organoids, the lowest concentration reducing cell viability following 7 days of exposure to MeHg was 10 µM [42]. Proliferating NSCs of rodent origin exposed to MeHg 25 nM or higher concentrations for >24 h activate caspase and calpain-dependent pathways [43,44,45,46]. Similarly, human NSCs derived from umbilical cord stem cells undergo apoptosis after exposure to 50 nM MeHg for 48 h [14]. Exposure to 25 nM MeHg for 24 h induced cell death in human fetal or embryonic NPCs [47,48,49] and exposure to 50 nM for 24 h was reported to alter mitochondrial biogenesis and increase ROS production in human cortical neural progenitor cells (ReNcell CX) [48]. However, exposure to higher concentrations of MeHg (250 nM for 24 h) [44] was required to induce apoptosis in C17.2 cells (mouse neural progenitor cell line). Interestingly, exposure to either 100 or 500 nM MeHg for 16 h [50] in adult NSCs isolated from male or female mice demonstrated sex-specific effects, namely that cells derived from females were less sensitive than cells from males. Caspase-dependent apoptosis induced by exposure to 100 nM MeHg for 48 h in mouse NSCs can be prevented by antioxidant treatment (NAC or alpha-tocopherol), and promoted by the inhibition of GSH synthesis [51]. Similarly, MeHg-induced cell death can be reduced by caspase [44,52] and calpain [44] inhibitors. Three-dimensional cell culture systems require a considerably higher concentration of MeHg or PFOS/PFOA to elicit similar effects as in the 2D culture of NSCs (see Table 1). Embryoid bodies appear less sensitive to MeHg than a monolayer culture of cells, and cell death has been documented after exposure to >100 nM for 14 days [53], 200 nM for 11 days [54], or 1000 nM for 16 h [55].

Exposure for 48 h to 100 nM PFOS showed a significant increased percentage of apoptotic nuclei only in primary cultures of rat cortical NSCs [56]. The analysis of cytotoxicity in C17.2 cells exposed to PFOS in the range of 25–200 nM for 48 h demonstrated a dose-dependent decrease in the number of cells [57]. In contrast, 3D cultures of mESC did not show a significant morphological change after 7-day PFOS exposure at a concentration of 10 µM [58]. hiPSC exposed to PFOS or PFOA (≥50 nM) for 24 h did not exhibit significant alterations in cell viability, but alterations in cell-specific differentiation, such as pancreatic, endocrine, and cardiomyocyte differentiation [59,60]. Cell viability and ROS generation assays suggested that PFOS did not induce cytotoxicity in mESCs at the concentrations tested (≤10 µM) for 7 days, but disrupted the expression of neural developmental [45,61] markers without affecting the proliferation of the differentiating cells [58]. Similarly, rat hippocampal NSCs exposed to PFOS or PFOA (≥200 nM) for 24 h did not show significant morphological changes and PFOS even increased NSC viability [62], suggesting that cell culture conditions may modulate PFOS or PFOA effects.

### 4.2. Proliferation

Exposure to sub-toxic concentrations of MeHg does not reduce cell viability but it decreases cell proliferation. Monolayer cultures of primary NSCs derived from rat embryos (E14.5) undergo cell cycle arrest following 48 h exposure to MeHg at ranges from 2.5 nM to 10 nM [45,50]. Recently, Yuan and colleagues reported that 2D culture of murine NSCs derived from E12 exhibited a significant decreased proliferation after exposure to 0.25 nM, but not at higher doses (0.5–5 nM) [63]. In addition, MeHg-induced reduction of proliferation was associated with an upregulation of GSK-3b and CDK inhibitors p16 and p21; a significant increase in cyclin E degradation; and an alteration of cytoskeleton dynamics [63]. NSCs exposed to MeHg go through cellular senescence, as shown by the alteration of Bmi, Hmga1, and Hp1g gene expression [61]. Similar effects were shown in human NPCs (ReNcell CX) after exposure to 10 nM MeHg for 24 h, which were linked with the upregulation of p16, p21, and p53 [49].

NSC proliferation has also been evaluated after exposure to PFOS by staining with EdU, a thymidine analogue. Cells exposed to 25 or 50 nM PFOS for 48 h showed a significant decrease in the number of EdU-positive cells as compared to control cells [56]. The analysis of cell proliferation using the CCK-8 assay indicated that 50 nM of PFOS impaired the proliferation of C17.2 cells in both a time- and concentration-dependent manner [57]. This study also showed that the down regulation of the GSK-3β/β-catenin axis, and its target genes, cyclin D1, C-myc, Cox-2, and survivin, played a crucial role in the PFOS-induced reduction of cell proliferation. In contrast, the proliferation of NSCs derived from rat embryonic hippocampus has been shown to be increased after exposure to PFOS doses ranging from 1µM to 10µM for 48 h exposure. Conversely, PFOA exposure induced no alteration in NSC viability or proliferation [58,62]. mESC exposed to 10 µM PFOS for 7 days did not show the altered intensity of alkaline phosphatase (AP) staining compared to the control, meaning that self-renewal ability was not affected.

### 4.3. Differentiation, Migration, and Neurite Outgrowth

A study using differentiating C17.2 cells for mRNA expression profiling using microarrays with genome-wide coverage identified a set of 30 mRNA species strongly associated with neuronal differentiation, out of which 14 displayed significant alterations after exposure to 90 nM MeHg for 10 days [64]. The decrease in the number of neurons and altered neurite outgrowth confirmed the pattern of alterations suggested by the mRNA expression profile [64]. In NSC culture, the absence of growth factors (FGF or EGF) induces spontaneous differentiation. Rat NSCs exposed to MeHg at ranges from 2.5 nM to 10 nM for 48 h exhibited reduced neuronal differentiation [44,63,65,66]. Tian et al. reported that a sub-toxic concentration of MeHg (10 nM) induced alterations of hippocampal neurons and astrocyte differentiation that could be reversed by antioxidant treatment with polysaccharides from *Lycium bararum* [66]. These results suggest that oxidative stress may regulate NSC differentiation. Exposure to 10 or 25 nM MeHg for up to 12 days in human NSCs was shown to decrease neuronal differentiation [47,67,68]. Interestingly, Yuan et al. demonstrated that NSCs isolated from mouse embryos exposed to 0.25 nM for 3 days had an enhanced neuronal differentiation but a reduced number of precursor cells [63]. Similarly, exposure to 10 nM MeHg for 4 days reduced hNPCs neuronal differentiation that was associated with decreased expression of BDNF [47]. Using embryoid bodies derived from mouse ESC, Theunissen et al. [69] have shown that exposure to a subtoxic concentration of MeHg (25 nM for 8 days) reduced neuronal outgrowth and was accompanied by an upregulation of late neuronal differentiation genes and a downregulation of early differentiation genes. We reported that MeHg-induced alteration of NSC differentiation is mediated by ERK 1/2 dephosphorylation and Notch signaling pathways. In the same model, we also demonstrated that MeHg-induced effects on neuronal differentiation could be rescued by the metalloprotease inhibitor GM-6001, which prevented cleavage of the Notch receptor [65]. In addition to neuronal differentiation, exposure to 10 nM MeHg for 2 days increased astrocytic differentiation in human NES cells, which could be reversed by DAPT, a gamma-secretase inhibitor that blocks extracellular Notch cleavage [68]. Interestingly, the positive correlation of MeHg-induced alterations of astrocyte differentiation with NES cells derived from a patient with a mutation in the NRXN1 gene linked to autism spectrum disorder [68].

Human cells exposed to a range of concentrations from 1 to 50 nM of MeHg show decreased neuronal migration and neurite outgrowth [47,70,71,72]. Remarkably, MeHg-induced alterations were enhanced in cells of male origin [47]. Similarly, exposure to 500–1000 nM MeHg for 48 h reduced neuronal migration in the 3D (neurospheres) culture of human NPCs [6,73], and the effect was associated with a reduction in ERK 1/2 phosphorylation. Neurospheres generated from hNPCs exposed to MeHg for 3 days displayed reduced neuronal migration from 100 nM [74]. These findings are supported by the altered neuronal migration and positioning of cerebrocortical neurons following in vivo administration of MeHg (0.1 or 1 mg/kg/day i.p. GD11-21) [75]. However, no change in the proliferation or differentiation of NSC has been reported in the same experimental model [75].

In rat NSCs we have shown that exposure to PFOS at doses ranging from 25 to 50 nM for 48 h neuronal differentiation increases while the proportion of nestin-positive cells decreased [56]. We also demonstrated that nanomolar concentrations of PFOS increase neurite outgrowth and significantly increase the number of CNPase-positive cells (oligodendrocytes), whereas astrocyte differentiation is not changed [56]. In contrast, Pierozan and colleagues [62] demonstrated that exposure of embryonic hippocampal NSCs to 10 µM of different PFAS for 24 h led to altered neuronal cell body morphology, but had no effects on neurite number and length, or on the number of branches per cell [62]. In addition, they found no alteration in oligodendrocyte and astrocyte differentiation in NSCs exposed to PFAS. This result may depend on cell type and culture conditions. Using monolayer culture of mESC, Yin et al. demonstrated that exposure to 1 nM up to 10 µM PFOS for 9 days exerted a general inhibitory trend in a dose-dependent manner on the expression of the pluripotency marker Nanog and on neural marker genes such as Sox1, Sox3, Nestin, Pax6, and Map2 [58]. Similarly, in 3D cultures of mESCs, exposure to PFOS for 9 days affected the differentiation process [58].

### 4.4. Synaptogenesis and Network Formation

Mature, terminally differentiated neurons display spontaneous electrical activity and engage in the formation of synapses and networks in culture. Altered synaptogenesis as a neurotoxic effect can be illustrated by changes in the number and morphology of synapses based on the presence of postsynaptic proteins, such as PSD95 or neurotransmitter receptors; alternatively, it can be illustrated by presynaptic markers such as vesicular transporters for glutamate, vGLUT-1, or GABA, vGAT. In a recent study, synaptogenesis has been highlighted as the most sensitive endpoint by mathematical modeling of neurotoxicity in hiPSC-derived neurons [72]. Neurite outgrowth and the expression of synaptic markers were decreased following exposure to as low as 0.25 nM MeHg for 3 days or 0.05 nM MeHg for 14 days [72].

Both PFOS and PFOA have been shown to increase neuronal excitability following acute exposure (10 or 100 µM), but inhibit synaptogenesis and synaptic signaling upon chronic exposure (10 µM) in primary hippocampal neurons [76] and differentiated hiPSC [77]. In vivo, administration of the equivalent of 21 micromol/kg bodyweight PFOS or PFOA (11.3 and 8.7 mg/kg bodyweight, respectively) at postnatal day 10 increased the levels of synaptophysin in the cerebral cortex and the hippocampus, 24 h after administration [78]. The long-term effects in vivo, as well as the effects on synapse formation and function in DNT models in vitro remain to be investigated.

The evaluation of neuronal function, however, has received less attention, and the functional implications of impaired synaptogenesis are unclear. MeHg and PFOS/PFOA have been shown to alter intracellular calcium homeostasis and neurotransmitter receptor activity (reviewed in [39,79]). In recent years, multielectrode arrays (MEA) have been used to record electrical activity in large populations of neurons, including action potentials (“spikes”) and groups of action potentials (“bursts”) and network bursts. Dingemans and colleagues demonstrated in primary cultures of rat cortical neurons that exposure to 0.1 µM MeHg for 14 days did not affect cell viability but decreased neuronal firing (Spikes) [80]. Similarly, neuronal firing was inhibited by exposure to 1µM MeHg for 30 min, without reducing cell viability. In addition, the mean burst rate (MBR) was decreased in hiPSC-derived Glutaneuron-Astrocyte co-culture exposed to 30 µM MeHg for 30 min [81]. In the same experimental model, exposure to PFOS (0.1 µM) or PFOA (1 µM) for 30 min significantly decreased mean spike rate (MSR) and mean network bursts (MNB), while MBR was decreased by exposure to 100 µM PFOS or PFOA [81]. Cell culture type and conditions may contribute to variability in response to chemical exposure. In a recent experiment, Tukker et al. have shown that 30 min exposure to 100 µM PFOS increased network activity in primary rat cortical neurons, but decreased neuronal activity in hiPSC-derived neurons [77]. Relevant for neuronal differentiation, MEA systems allow longitudinal observations to capture a clear picture of functional alterations induced by exposure to potentially neurotoxic chemicals.

## 5. Molecular Mechanisms Associated with NSCs Dysregulation

### 5.1. Oxidative Stress and Mitochondrial Impairment

Induction of oxidative stress is the most common mechanism and appears to be critical in NSCs undergoing toxic exposures (see Table 1 and Table 2) [12,48,49,66,82]. Compelling evidence shows that MeHg-induced reactive oxygen species (ROS) formation and impaired mitochondrial function, as shown by in vivo and in vitro studies [83]. Mitochondria contain different antioxidants, including glutathione (GSH), thioredoxin (TRX), and the catalase system, which quench ROS and maintain an oxidant-antioxidant balance. MeHg is a threat to the antioxidant defenses, which further alter the REDOX balance essential for proper mitochondrial function. Based on our data from NSCs, exposures to high levels of MeHg (>10 nM for 48 h) damage mitochondrial functions with the release of cytochrome *c* and activation of the caspase-dependent apoptotic cell death pathway [44]. While subtoxic concentrations of MeHg (<5 nM for 48 h) do not affect cell survival [61], the expression of genes of mitochondrial respiratory chain enzymes of complexes I and III is repressed. Antioxidants protect from MeHg-induced damage, preventing both apoptosis [51] and the alterations in neuronal differentiation [66] in NSCs isolated from embryonal rodent brains exposed to low-dose (10 nM) MeHg.

Alterations of different genetics and epigenetics pathways are associated with oxidative stress that generates both free radicals and nonradical oxidants. Free radicals give rise to macromolecular damage and nonradical oxidants (e.g., H_2_O_2_, peroxynitrite, lipid hydroperoxide, and disulfides) disrupt redox signaling pathways. Hydroxyl (•OH) is a free radical that can react with guanosine directly, oxidizing it to 8-oxo-7,8-dihydro-2 deoxyguanosine (8-oxo-dG). While 8-oxo-dG is usually repaired by base excision repair (BER) mechanisms, it can give rise to G/T transversions (point mutation) by mispairing with adenine instead of cytosine (see Figure 2). Such mutations in mitochondrial DNA have been demonstrated in human NPCs (ReNcell CX) exposed to 10 or 50 nM MeHg for 48 h [48].

The molecular mechanisms of PFOS-induced neurotoxicity remain largely obscure. PFOS has been shown to decrease cell viability in human-derived neuroblastoma cells (SH-SY5Y) by increasing ROS [84], but it is not clear whether PFOS-induced oxidative stress affects NSC viability. Activation of the JNK pathway and accumulation of ROS have been demonstrated in PFOS-exposed C17.2 cells, which suggests a link between ROS production and JNK signaling that may critically contribute to PFOS-induced neuronal apoptosis [57]. PFOS-induced activation of JNK signaling results in the expression of pro-apoptotic proteins and the initiation of the mitochondrial apoptotic pathway. In addition, JNK may also promote the expression of both pro-apoptotic Bcl-2 family proteins to trigger mitochondrial apoptotic cascades and alter Nrf2 expression that may regulate the expression of antioxidant proteins that protect against oxidative damage triggered by injury and inflammation [84]. ROS are crucial players in neuronal death under various pathological conditions, and may directly execute cell death by inducing mitochondrial permeability and releasing cytochrome *c*, or initiating multiple signaling pathways, such as p53-p21, JNK, and FOXO to trigger neuronal death [85]. In line with the above evidence, we showed that PFOS induces alterations of the mRNA expression of PPARs (PPARα, PPARδ, and PPARγ) and their downstream targets; the mitochondrial uncoupling proteins (UPC1, UCP2, and UCP3), and the superoxide dismutase (SOD1, SOD2, SOD3) which are important enzymes in the antioxidant defense system of primary culture of embryonic cortical NSCs [56,86]. We also showed that PFOS induces upregulation of PPARγ and UCP2 associated with the accumulation of ROS and oxidative stress. Thus, these results indicate that PFOS-induced ROS production may be a primary neurotoxic mechanism. In contrast, there is no evidence for oxidative stress to contribute to the neurotoxic effects of PFOA in NSCs.

**Table 1 ijms-24-04338-t001:** Studies reporting the effects of MeHg exposure in NSCs.

Exposure Doses (nM)	Duration of Exposure	Model	Outcome (LOAEC or IC5)	Analyzed Markers	Ref.
NSCs of mouse origin					
0.25, 0.5, 5	3 days	NPCs (E12–14.5)	Decreased proliferation; increased neuronal differentiation (0.25 nM) Decreased differentiation (0.5 nM)	Ki67, Nestin, Sox2 Tuj1	[63]
2.5, 25, 200, 250	3–13 days	ESCs (D3); EB	Decreased neurite outgrowth (2.5 nM) Increased % cells in an undifferentiated state (250 nM) Induced cell death (200 nM)	% EB with >75% outgrowth *Nestin, SSEA1* Alamar Blue cytotoxicity assay	[54]
25	8 days	ESCs (D3); EB	Downregulation of transcription and development-related genes; upregulation of neurodevelopment- and cell motion-related gene sets Decreased neural outgrowth	Whole-genome transcriptomics analysis % EB with >75% outgrowth	[69]
15.13–1000 160–80,000	8–14 days 16 h	ESCs (D3)	Decreased neuronal differentiation and increased astrocyte differentiation (62.5 nM) Induced cell death (5 uM)	Nestin, Tuj1, MAP-2, GABAA-R, GFAP; MTT assay	[55]
0.1–10,000	14 days	ESCs (D3); EB	Altered gene expression-related neuronal differentiation (0.1 nM) Decreased cell viability (200 nM)	Nestin, Pax6, Tuj1, NCAM1, Nefm, MAP-2 *(Mtap2)*, Resazurin reduction	[53]
100, 500	16 h	adult NSCs	Induced apoptosis (males more affected than females) (100 nM)	Condensed nuclei (Hoechst 33342)	[50]
100, 1000	48 h	NPCs (E14.5); EB	Induced apoptosis (prevented by antioxidant treatment) (100 nM)	TUNNEL staining	[51]
100, 500, 1000	3–24 h	NPCs (E14.5); EB	Induced apoptosis (prevented by caspase inhibitors) (100 nM)	Cleaved Caspase 3, Bcl-2, Bax, DNA fragmentation (ladder)	[52]
90	10 days	C17.2	Reduced neuronal differentiation; reduced number of neurites	Genome-wide microarray	[64]
NSCs of rat origin					
2.5–50,000	48 h	NPCs (E13)	Reduced cell proliferation (2.5 nM) Decreased cell viability and induced apoptosis (500 nM)	Cyclin E Caspase 3 activation	[43]
0.3–10,000	6–24 h	NPCs (14.5)	Reduced cell viability and proliferation (3 µM @6 h; 300 nM @24 h)	Cyclin E; thymidine incorporation	[46]
1.0–1000	48 h	NPCs (E14)	Reduced cell viability and proliferation (10 nM)	Cyclin D1, Cyclin E and CDK2	[45]
2.5–10	6 h–7 days	NPCs (E15)	Inhibited neuronal differentiation (2.5 nM)	Notch, Tuj1	[65]
2.5–5.0	24–48 h	NPCs (E15)	Inhibited neuronal differentiation (2.5 nM) Reduced cell viability and induced apoptosis (25 nM)	DEVD-AMCC leavage, Bax, CytC,	[44]
2.5–5.0	48 h	NPCs (E15)	Reduced cell proliferation Senescence	Cell cycle regulating genes (*p16, p21*), mitochondrial genes (*Nd3*, *Cytb*), senescence (*bmi*, *Hmga1,* Hp1ɣ)	[61]
2.5–5.0	48 h	NSCs (E16)	Inhibited neuronal differentiation (2.5 nM) Increased astrocyte differentiation (2.5 nM)	MAP-2, GFAP	[66]
NSCs of human origin					
10, 50	24 h	NPCs (ReNcell CX cells)	Oxidative stress (10 nM). Decreased cell proliferation (50 nM)	Cell cycle regulatory genes (*p16, p21, p53*); ROS production	[49]
30–3000	5 or 10 days	H9-derived hNPC	Decreased cell viability (IC5, 10 nM 5 days; 100 nM 10 days) Decreased cell migration (IC5, 100 nM 10 days)	CellTiter Blue ^®^ viability assay; neurite area	[40]
10, 25, 100, 25 3–3000	72 h	hNPCs (3D life Hydrogels); hNPCs, (neurospheres)	Decreased migration distance (100 nM)	Cell morphology after plating in 2D (maximum extension)	[74]
0.6–10,000	14 days	NSCs derived from MR90-hiPSCs	Decreased cell viability (IC5, 50 nM) Decreased synaptogenesis, neurite outgrowth, and BDNF expression (50, 130 nM)	SYP, PSD95, MAP2, beta-III-tubulin, BDNF	[72]
30–3000	24 h	hiPSC-derived hNPCs and primary hNPCs (neurospheres)	Decreased neuronal migration (300 nM) Decreased viability (1000 nM)	Maximum extension; Cell Titer-Blue^®^ Viability Assay	[41]
250–1000	48 h	hNPCs (neurospheres)	Increased neuronal differentiation (250 nM) Decreased neuronal differentiation (750 nM) Decreased neuronal migration (500 nM)	Neurosphere morphology; GFAP, B-III-tubulin	[6]
2.5–12.5	48 h	hiPSC-derived neuroepithelial stem (NES)	Increased astrocyte differentiation (10 nM)	GFAP	[68]
2.5–100	4 days	hiPSCs	Decreased neuronal differentiation (10 nM) Increase cell death (25 or 100 nM)	Tuj1, BDNF and CDKL5	[47]
0.5–50	48 h	hESCs (H9-derived neural crest cells)	Decreased neuronal migration (10 nM)		[70]
10–50	24 h	ESCs (ReNcell CX cells)	Reduced cell viability; induced apoptosis (50 nM) Decreased mitochondrial functions (50 nM) Increased ROS (10 nM)	mitochondrial genes (*ND1, Cytb, ATP6)*, Mitochondrial membrane potential (JC-1 aggregation), ROS production	[48]
100–10,000	1 week	hiPSC organoids (BrainSphere)	Decreased cell viability (10 µ M) Decreased myelination (10 µ M)	MBP; neurofilaments; PLP1	[42]

**Abbreviations:** CDK2, cyclin dependent kinase 2; MAP-2 (Mtap2), microtubule associated protein 2; Nd3, mitochondrially encoded NADH dehydrogenase 3; Cytb, mitochondrially encoded cytochrome b; Bmi1, polycomb ring finger oncogene; Hmga1, high mobility group AT-hook 1; Hp1ɣ, heterochromatin protein 1; EB, embryoid body; ND1, mitochondrially encoded NADH dehydrogenase 1; ATP6, mitochondrially encoded ATP synthase 6; LOAEC—lowest observed adverse effect concentration; IC5—concentration corresponding to very low biological effects, estimated from the dose-response curve.

**Table 2 ijms-24-04338-t002:** Studies reporting on the effects of PFOS or PFOA in NSCs.

Exposure Doses	Duration of Exposure	Models	Alterations (LOAEC)	Analyzed Markers	Ref.
12.5–200 nM PFOS	1–72 h	Mouse-derived C17.2 cells	Decrease cell viability and proliferation (25 nM)	Ser9 phosphorylation, c-myc, cyclin D1, and survivin	[57]
25–50 nM PFOS	48 h	Rat embryonic cortical NSC	Increased apoptosis (50 nM) Decreased proliferation. Increased neuronal and oligodendrocyte differentiation. Decreased Ca2+ activity (25 nM).	TuJ1, GFAP, CNPase PPARγ, PPARα or PPARδ, UCP2, UCP3	[56]
0.1–10 µM PFOS	9 days	Mouse embryonic stem cells (mESC)	Altered cell viability or self-renewal abilities of mESCs (100 nM). Decreased neuronal differentiation (100 nM)	Sox1, Sox3, nestin, pax6, and Map2	[58]
1–250 µM PFOS 1–250 µM PFOA	24 h	Rat, hippocampal NSCs (E15)	Increased cell proliferation Decreased cell body area Altered cell body area at and the neurite network of the NSC-derived neurons	Formazan production, *Nestin,* Tuj1, GFAP, Oligo4,	[62]

### 5.2. Epigenetic Alterations

Epigenetic alterations are defined as chemical modifications of DNA that are complementary to the genetic code for turning genes on or off via chromatin remodeling without changing the DNA sequence. Epigenetic mechanisms include (1) histone modifications (the modulation of DNA availability via condensation or relaxation of chromatin wrapping in nucleosomes for binding transcription factors); (2) DNA methylation, which modulates the availability of DNA for binding transcription factors in the DNA transcription machinery (gene expression is suppressed when DNA is methylated in the promoter region); and (3) non-coding RNA, such as microRNA (miRNA) strands, which target sequences in the mRNA and repress gene expression post-transcriptionally. Epigenetic alterations describe a variety of reversible modifications across cell types in an organism, which regulates a wide range of physiological and pathological processes from the meiotically and mitotically cell cycle to the function of non-dividing cells—such as neurons. A wide range of exogenous factors, such as the availability of methyl donors in the diet, and environmental contaminants, influence the modification of epigenetic marks. In recent years, the association between oxidative stress and epigenetic alterations has been suggested as a potential mechanism of environmentally relevant exposure to toxicants.

#### 5.2.1. DNA Methylation

DNA is methylated by the enzymatic family of DNA-methyl transferases (Dnmts), which add a methyl group to cytosine in position 5 to generate methylcytosine (5-mC). This 5-mC is oxidized by a family of Ten-eleven translocation methylcytosine dioxygenases (Tets) and generates 5-hydroxymethylation (5-hmC), and the demethylation process is completed during DNA replication. In addition, 5-hmC can be further oxidized by Tets to produce 5-formylcytosine (5-fC) and 5-carboxylcytosine (5-CaC) before demethylation by replication. Oxidative stress may interfere with DNA methylation in two ways. First, it reduces the availability of methyl groups required for DNA methylation by reducing the activity of, e.g., methionine-adenosyltransferase and methionine synthase, enzymes catalyzing the synthesis of S-adenosylmethionine (SAM; the main donor of methyl groups for DN methylation). Second, ROS can oxidize DNA and generate 8-oxo-dG, which activates DNA base repair enzyme OGG1. Thus, activation of OGG1 prevents DNMTs from methylating the DNA, but recruits Tet1 to demethylate the adjacent 5-mC directly or indirectly via deamination followed by Base Excision Repair (BER) enzymes during replication. When this BER cannot repair the specific base, it will be replicated as a point mutation (see Figure 2).

Wang and colleagues showed a significant increase in ROS production in hNPC exposed to low levels of MeHg (10 or 50 nM) [48]. Similarly, we demonstrated that in the primary culture of embryonic rat NSCs low levels of MeHg (2.5 or 5 nM for 48 h) induce long-lasting effects which are still present in cells that were not directly exposed to MeHg and had levels of Hg below the detection limit [61]. Interestingly, the Hg concentrations measured in NSCs exposed to 2.5 and 5 nM were 0.4 and 0.7 ppm respectively, which are comparable to the concentrations reported in *post-mortem* material from infants exposed to MeHg from the maternal diet (up to 0.3 ppm; [87]). The upregulation of p16 and p21 that we detected in NSCs was associated with decreased proliferation (senescence) [61]. We observed an association between global DNA hypomethylation and Dnmt3b downregulation. This is in line with an earlier study showing that inhibition of DNMT decreases cell proliferation and induces cellular senescence in HUBC-NSCs associated with the upregulation of p16 and p21 [88]. Go et al. reported that LUHMES (CRL-2927) cells exposed to 1 nM MeHg for 6 days had increased global DNAA methylation, associated with DNMT1, 3A, and 3B, and following in vivo exposure (3 mg/kg/day between GD12 and GD14) obtained similar results [71]. These apparently contradicting results can be explained by the experimental protocols that lead to the accumulation of Hg to levels relevant for massive and accidental exposure during development. Furthermore, experimental models of aging demonstrated a link between oxidative stress and DNA demethylation [89]. A recent study showed that prenatal exposure to MeHg via maternal diet affected gene-specific methylation, important in brain development, and neuronal signaling in 7-year-old children [90]. CpG sites in the promoter regions of NR3C1 (glucocorticoid receptor), GRIN2B (NMDA-receptor subunit), and BDNF (neurotrophic factor modulating neuronal development and function) were hypermethylated following the developmental exposure to MeHg. For NR3C1, MeHg-induced de novo methylation was found in CpG3 and CpG5 sites, where CpG3 is part of the binding site for transcription factor Hen-1. CpG3 and CpG4 are the binding sites for the transcription factor NGFI-A. Thus, CpG3 methylation downregulated NR3C1 by inhibiting NGF1-A binding. However, MeHg-induced de novo methylation in CpG4 did not affect NR3C1 expression but downregulated GRIN2B expression by inhibiting nuclear respiratory factor 1 (Nrf1). Notably, neurodevelopmental disorders, such as ADHD, ASD, and schizophrenia have been associated with functional alteration of GRIN2B. Similarly, MeHg-induced CpG5 decreased BDNF expression, which is essential for neuronal development, nerve cell survival, and synaptic plasticity [91]. In addition, BDNF is a particularly relevant target of MeHg because BDNF polymorphisms increase the susceptibility to neurotoxicity and MeHg-induced BDNF downregulation has been associated with depression [92,93].

There is currently no evidence that PFAS changes DNA methylation in NSCs at doses relevant to human exposure. Several studies have reported that global and gene-specific methylation alterations are associated with a micromolar concentration of PFOS or PFOA exposure (100–400µM) in various human cell lines and blood samples [94,95,96]. These studies indicate that PFAS may also change the epigenome of NSCs. Further evidence showed that PFOS could decrease global DNA methylation and methylation of the LINE-1 regulatory region, but increase the GSTP promoter region methylation. Therefore, PFOS could lead to the CpG methylation of BDNF mediated by DNMTs and decrease the expression of BDNF. This study explored the mechanism by which PFOS affected BDNF expression via miRNA and methylation regulation. In addition, PFOS exposure decreases the expression of BDNF at mRNA and protein levels, increases the expression of microRNA-16, microRNA-22, and microRNA-30a-5p, decreases the expression of DNMT1 at mRNA and protein levels, but increases the expression of DNMT3b at mRNA and protein levels [97]. It has also been shown that PFOS exposure changes the methylation status of BDNF promoters I and IV. These findings suggest that the downregulation of BDNF along with the upregulation of BDNF-related microRNA might underlie the mechanisms of PFOS-induced neurotoxicity [97].

#### 5.2.2. Histone Modifications

Histone modifications of epigenetic mechanisms regulate gene transcription via covalent posttranslational modifications (PTMs), such as acetylation, methylation, phosphorylation, ribosylation, ubiquitination, sumoylation, or glycosylation. Both histone acetylation and methylation marks are redox-sensitive and heritable. Approximately 30 histone acetyltransferases (HATs) and histone deacetylases (HDACs) have been found to regulate histone acetylation. Therefore, gene transcription depends on the balance between HATs and HDACs applied to the epigenetic marks. Methylation of histone can either upregulate or downregulate gene expression, which depends on the position of the histone tail, and the number of methyl groups added to a particular amino acid (lysine or arginine) for methylation. More than 40 histone methyltransferases (HMTs) and demethylases regulate this dynamic process. We showed earlier that in utero exposure to 0.5 mg/kg/day MeHg from GD7 until PND7 induces depression-like behavior in male mice [92] as well as decreased granule cell proliferation in the hippocampal dentate gyrus [61]. In this model, the Hg concentration found in the mouse brain was about 0.9 ppm [93], in the same order of magnitude as the one measured in humans [87]. We demonstrated that MeHg increased histone H3-K27 tri-methylation, and decreased H3 acetylation at the BDNF promoter IV region, which resulted in the downregulation of BDNF expression. [61]. Decreased BDNF has been linked to the onset of depression, and MeHg-induced depression in mice is reversed by the antidepressant fluoxetine, which increases the levels of BDNF [92]. These findings may be relevant for the possible effects of MeHg on human mental health.

#### 5.2.3. Non-Coding RNAs

Noncoding RNAs, such as micro-RNAs (miRNAs), long noncoding RNA (lncRNA), and circular RNA (circRNA) are epigenetic processes involved in the post-transcriptional repression of specific genes. Pallocca et al. [98] have used differentiating NT2 cells to evaluate the use of microRNA profiling as a biomarker for developmental neurotoxicity. Exposure to 400 nM MeHg for 5 weeks resulted in the overexpression of a cluster of 5 microRNA species (miR-302b, miR-367, miR-372, miR-196b, and miR-141). The analysis of mRNA expression for genes known to be regulated by these microRNAs identified alterations in line with cognate effects of MeHg, such as decreased neuronal differentiation, and cellular stress response [98]. These findings, however, need to be replicated in NSCs in order to validate the use of microRNA regulation signatures for DNT studies. Recent evidence demonstrated that MeHg-induced alteration of cell viability and decreased cell proliferation were associated with the upregulation of p53R2 expression [49]. In this study, they demonstrated a negative correlation of different miRNAs (miR-1285, miR-30d, and miR-25) with p53R2 expression. While it is not clear whether miR-25 regulates p53 expression directly, overexpression of this small RNA in MeHg-treated ihNPCs significantly reduces the protein expression of p53 [49]. Thus, this study indicates diverse mechanisms of MeHg-induced developmental neurotoxicity.

### 5.3. Indirect Neurotoxicity and Intrinsic Limitations

In this section, we have focused on outcomes and mechanisms of direct neurotoxicity. In vivo data point to biologically relevant effects on neurophysiology and behavior due to indirect neurotoxic effects. Endocrine disruption is particularly relevant during development, as acknowledged in the following definition: “Developmental neurotoxicity (DNT) refers to any adverse effect of perinatal exposure to a toxic substance on the normal development of nervous system structure and/or function.” [99]. During prenatal development, indirect neurotoxic effects due to interactions with the placenta or maternal organs should also be taken into consideration (see [100,101]). The toxic effects of exposure to MeHg are largely restricted to the nervous system, but recent evidence points to potential endocrine-disrupting effects on glucocorticoid receptor signaling [102,103]. For PFAS, the investigation of endocrine disruption in relation to developmental neurotoxicity has focused mainly on the thyroid hormone system, but altered signaling via gonadal and stress steroids may also have a biologically relevant contribution (reviewed in [39,101]).

In this context, one of the main limitations is that NSC/NPC models can only provide information on direct neurotoxicity, while interactions with other organs and systems remain difficult to investigate. Indirect neurotoxicity, including but not limited to endocrine disruption, as well as compensatory mechanisms acting at later developmental stages may account for discrepancies between neurotoxic outcomes predicted by NSC models and in vivo or epidemiological observations. The development of 2D and 3D systems for neural stem cell cultures has advanced our understanding of neurodevelopmental processes in physiological conditions, as well as the effects of exposure to toxicants (see the evaluation of myelination in BrainSpheres [42]). Organoids offer the possibility to investigate the more complex processes affected by developmental neurotoxicants by recapitulating brain development in a dish, and the availability of cells of human origin increases the relevance of the results. Co-culture systems and, more recently developed, vascularized organoids [16] may further support the extrapolation of findings to human populations.

Additional intrinsic technical limitations stem from cell source and culture conditions. The embryonic developmental stage at the time of harvesting may have a significant impact on susceptibility to neurotoxicity. The study by Edoff et al. [47] indicates that human NPCs derived at earlier embryonal development stages are more sensitive to MeHg exposure. The same study also highlights the fact that the sensitivity to neurotoxic insults also differs according to the sex of the embryo, namely a more pronounced neurotoxic effect in male hNPCs [47]. There is limited information on sex-related differences from the reviewed literature (see e.g., [47,50]). For NSCs derived from animal models, the cells are typically harvested from several pups and pooled to increase the initial yield without specifically selecting male or female pups. The main constraint for NSCs of human origin is the availability of original material, and consistently reporting the sex and developmental age of the source of cells is critical for assessing the generalizability of the findings (see e.g., [74]).

## 6. Discussion and Perspectives

Even if the impact of exposure to such environmental contaminants has been decreasing thanks to increasing awareness and control of both industrial emissions and main sources of exposure [104], the estimated impact on healthcare expenditure remains considerable. MeHg is generated by anaerobic bacteria in water sediments and undergoes bioaccumulation and bioamplification in the food chain. PFOS and PFOA have relatively low acute toxicity but have quickly gained attention because of their pervasiveness and persistence in the environment. Human populations are continuously exposed through the consumption of contaminated food, water, and beverages. The neurodevelopmental impact of these environmental contaminants has been evaluated based on estimated intellectual disability and IQ points loss due to prenatal exposure. For MeHg, the annual cost has been estimated to be 2.84 billion USD [104]. For PFOS and PFOA, the annual increase in health expenditure attributable to loss of IQ points associated with low birth weight was estimated to be 1.11 billion and 10.7 billion USD for PFOA and PFOS, respectively [105]. The economic impact of other neurodevelopmental disorders associated with prenatal exposure to these toxicants (intellectual disability, ASD, ADHD, and possibly depression, as indicated by experimental data and occupational exposure [92,106,107]) would considerably increase the cost estimates. Therefore, new information on the mechanisms of action obtained from the most relevant models is necessary to identify, prevent and counteract their harmful effects.

The revised literature converges on NSCs (of rodent and human origin) being a valuable model for investigating developmental neurotoxicity. In vitro models seem to support observations made in epidemiological studies, such as higher sensitivity to toxicants in developing subjects (as in NSCs) than in more mature subjects (as in differentiated neurons and glia), as well as sex-related differences in susceptibility. Low levels of exposure induce major alteration in critical neurodevelopmental steps, which presumably leads to functional impairments. Nanomolar levels of relevant contaminants, such as MeHg and PFAS, alter proliferation, differentiation, migration, and neurite outgrowth, while micromolar concentrations induce apoptotic cell death. Neural stem cells of human origin (embryonal or iPSCs-derived), cultured in 2D or 3D systems, including organoids, are the best options available to explore normal neurodevelopment and its alteration in a controlled experimental environment. The fact that NSCs illustrate the neurotoxic potential of known neurotoxicants (such as MeHg) at low exposure levels that are relevant for the general population makes them an ideal in vitro model for identifying new neurotoxic chemicals and the effect of mixtures (see also [59,72]).

Altogether, the reviewed literature shows that mechanistic studies are important to support epidemiological and animal experimental data regarding the role of chemical contaminant exposures in the etiopathogenesis of neurodevelopmental disorders. The effects of MeHg and PFAS on proliferation and differentiation are similar in single-exposure models. Oxidative stress, Ca^2+^ homeostasis, and mitochondrial function impairments are common mechanisms behind cell death. Instead, the signaling pathways regulating NSC proliferation and differentiation affected by MeHg and PFOS/PFOA are different. The occurrence of epigenetic modifications enhances the complexity of the processes initiated by toxic exposures and highlights the risk of transgenerational effects. The identification of common and specific intracellular processes activated by exposures to neurotoxicants may enable the development of preventive and therapeutic strategies to benefit the populations at risk. More research is required to fill the knowledge gaps on most of the chemicals we are exposed to.

## Figures and Tables

**Figure 1 ijms-24-04338-f001:**
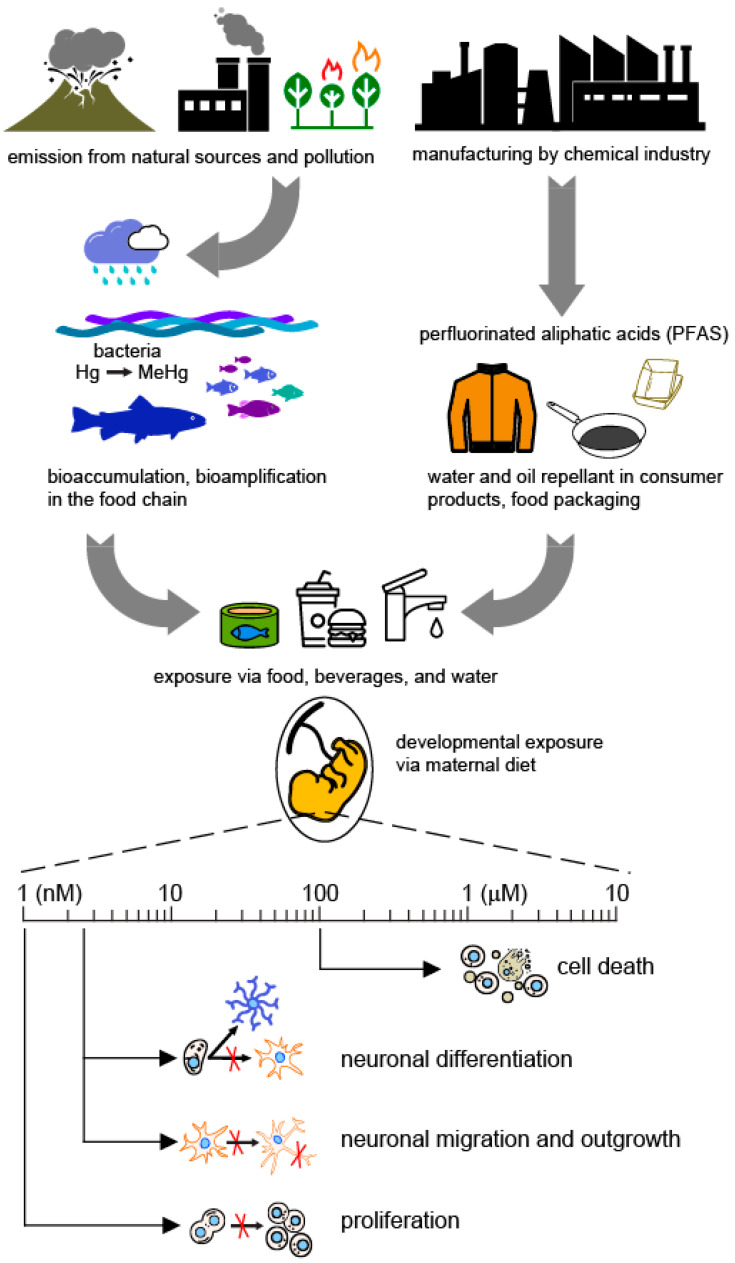
Overview of MeHg/PFAS exposure paths and toxic effects in NSCs. PFAS have been manufactured on a large scale and have been used in a wide range of consumer products. Inorganic mercury is released into the environment from natural and anthropogenic sources and enters the food chain after conversion to MeHg by sulphate-reducing bacteria in aquatic environments. Humans are mainly exposed via the consumption of contaminated water, beverages, and food. Both PFAS and MeHg cross the placenta, thereby posing a serious threat to the developing nervous system by the impact on fundamental neurodevelopmental processes. The concentration scale refers to the lowest concentrations of MeHg or PFAS reported to have an effect on NSCs.

**Figure 2 ijms-24-04338-f002:**
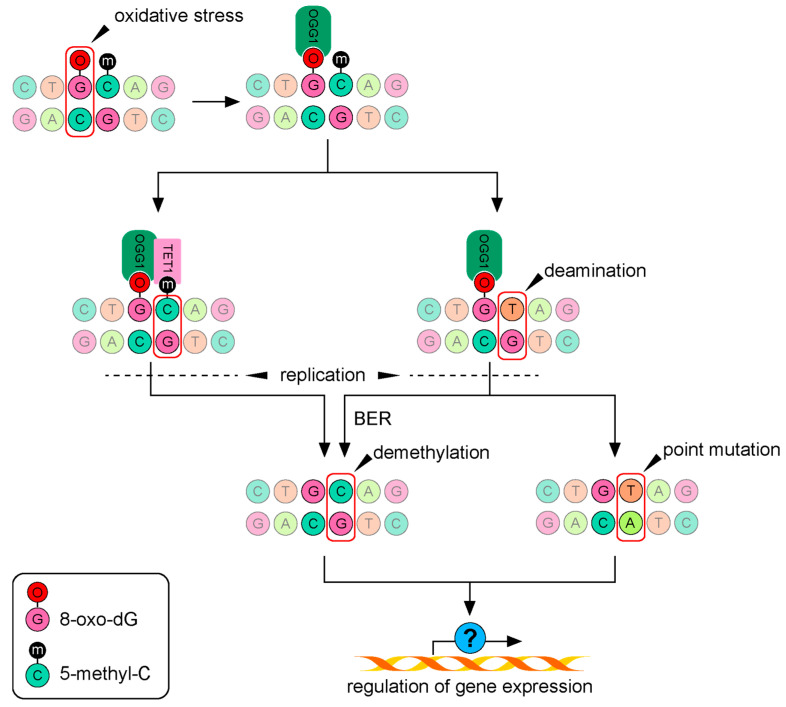
Oxidative stress can alter gene expression regulation by inducing both point mutations and epigenetic changes. Oxidized guanosine is recognized by 8-oxoguanine glycosylase (OGG1), a DNA base repair enzyme involved in base excision repair (BER). It can either recruit TET1 to bind and demethylate neighboring methylated cytosine (5-mC) or induce point mutations by initiating the deamination of 5-mC to thymine. During DNA replication, the thymine can be recognized and replaced with cytosine (BER), which removes DNA methylation. Alternatively, the point mutation is propagated by DNA replication, with unpredictable effects on gene expression.

## Data Availability

No new data were created or analyzed in this study. Data sharing is not applicable to this article.

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
