# Peer review of "The Threat Posed by Environmental Contaminants on Neurodevelopment: What Can We Learn from Neural Stem Cells?"

_ijms, 2023, doi:10.3390/ijms24054338_

Round 1
Reviewer 1 Report
The focus of the manuscript is to review neural stem/progenitor cell (NSC/NPC) models as tools to evaluate the threat of environmental contaminants on neurodevelopment. Methylmercury, PFOS and PFOA are reviewed examples of contaminants. However, the review lacks descriptions of essential events in neurodevelopment (i.e. synaptogenesis and neuronal networks), well-known mechanisms of developmental neurotoxicity (DNT) caused by endocrine disrupting chemicals and transcriptomic alterations caused by the contaminants. Hence, the review lacks a number of references, some are mentioned below.
Disruption of Ca2+ homeostasis, neurotransmitter signaling and neuroendocrine disruption are mentioned as neurotoxic mechanisms of PFAS (page 5, lines 220-221). However, nothing about these key events is elaborated in section 5. It is suggested that section 5 is extended by (at least) a subsection in which endocrine disruption of the toxics is reviewed.
In section 4, the use of NSC/NPC for studies on apoptosis, proliferation, differentiation, migration and neurite outgrowth are reviewed. The authors should also review studies of MeHg, PFOA and PFOS on synaptogenesis (eg. as section 4.4) and network formation (as section 4.5), to include most hallmarks of neurodevelopment in the review. If no such studies exist, this should be mentioned as knowledge gaps.
Page 7, section 5.1. Define what is meant by high and subtoxic concentrations in nM and indicate the dose in figures of the “low-dose” MeHg (line 354). In the same section, but on page 8, include the concentrations of PFOS that induce the effects seen. No effects of PFOA on oxidative stress and mitochondrial impairment are presented. Focus on effects studied in NSC and NPC.
Section 5.2. Epigenetic alterations are extensively reviewed, but it is not necessary to describe epigenetic modification in general. Focus on epigenetic alterations that are caused by the toxicants in NPC and NSC and how these effects are linked to human exposure and adverse outcomes. Effects in neuronal cell lines and cancer cells can be removed.
Page 11, line 531: How is depression linked to DNT? Give a reference.
The text in lines 535-548 on page 11 needs references. Indicate which compound is meant in line 541.
Table 1. Is incomplete regarding MeHg-exposed NSC/NPC. See references below under References. Indicate the models in more detail as in Table 2. Analyzed markers and reference are missing in bottom of page 2.
Table 2. Indicate which PFAS is indicated in each study.
Table 1 and Table 2. In addition to the exposure doses, indicate the dose/concentration that gives effect (eg. bench mark concentration or lowest observed effect concentration).
It would be nice to see section in which knowledge gaps are summarized and how these gaps can be filled by studies using NSC models.
Figure 1 is really nice and summarizes the review well. However, is the scale of concentration valid for MeHg, as well as for PFOA and PFOS?
Figure 2 does not add anything to the objectives of the review as presented in the abstract and the title.
References.
Although the lab of Ceccatelli is one of the most recognized in the field, the reference list is too selective with no less than 13 self-citations. On the other hand, important references from other groups are missing. Regarding NSCs/NPCs exposed to MeHg, papers by DeLeeuw et al., 2022; Chesnut et al., 2021; Pistollato et al., 2020; Hellwig et al., 2018; Hoffrichter et al., 2017; Attoff et al., 2017 and Palloca et al., 2013 are some examples of studies when iPSC, hNPC, NT2 cells and C17.2 cells have been used as models. Consequently, the findings in these papers should be included in section 4 and 5 and in the tables (see comments above).
Refs 5, 25 are incomplete
Ref 8 is missing journal
Refs 64, 74 and 80 are identical
Refs 72 and 73 are identical
Ref 77 is incorrect
Refs 86 and 87, doi numbers are missing
Minor:
Section 4.1. The first sentence should include the name of the toxicant. …. Higher concentration of what? MeHg?
Remove “®” from human ReN cells.
Add references to blood concentrations of MeHg (page 4).
Reviewer 2 Report
The authors of the manuscript do a thorough job of summarizing the threat posed by environmental contaminants on neurodevelopment: what can we learn from neural stem cells? The manuscript is well written and has taken into consideration all the recent literature that has been published in this domain.
· The authors should use the recently published papers and remove the old references (before 2010)
· 101 references for this paper are too low and the authors should use additional new references.

Round 2
Reviewer 1 Report
The authors have responded to the reviewers' comments and the manuscript is thereby significantly improved.